# The Current Contexts of Newly Graduated Nurses’ Competence: A Content Analysis

**DOI:** 10.3390/healthcare10061071

**Published:** 2022-06-09

**Authors:** Areum Hyun, Marion Tower, Catherine Turner

**Affiliations:** 1School of Nursing, Midwifery & Social Work, University of Queensland, St. Lucia, QLD 4067, Australia; m.tower@uq.edu.au; 2College of Nursing and Midwifery, Charles Darwin University, Casuarina, NT 0810, Australia; catherine.turner@cdu.edu.au

**Keywords:** competency, competence, new graduates, content analysis, Leximancer

## Abstract

Producing and sustaining a competent nursing workforce is imperative to protect the public. Identifying current issues and trends in nursing competence can strengthen insights and provide direction for the future nursing workforce. A summative content analysis was performed; PubMed, CINAHL, and Scopus were searched for content from the last ten years. A total of 3225 titles and abstracts regarding nursing competence were identified and analysed using the big-data analysis software Leximancer. Five themes were discovered from the analysis: (1) standardisation of nursing competencies with emerging competencies, (2) assessment competency levels, (3) graduates’ expectations and achievement, (4) safe and quality practice with teamwork, and (5) competency curriculum development. This found standardised nursing competencies, which suggests prioritising which core competencies should be focused on during education to produce competent generalist professional nurses, and employers could help nursing graduates improve their competence in specialised areas. This review also suggests that further education strategies should be developed to better prepare graduates for culturally safe practice to meet the needs of diverse minority populations and for informatics competency during the COVID-19 pandemic. Competence assessment methods must be extensively investigated to measure nursing competencies both longitudinally and cross-sectionally.

## 1. Introduction

Producing a competent nursing workforce is essential to meet population needs and protect the public [1]. Although higher education providers strive to produce nursing graduates who are well-prepared for their future jobs, employers continue to raise concerns about efficacy of nursing education programs in achieving this, stating that graduates are often not ready for the realities of the workplace [2,3]. Ensuring new graduates are competent is considered particularly important in influencing directly patient safety and quality care [4]. However, global concerns persist about a perceived disconnect between educational preparation to become a registered nurse and future employers’ expectations about new graduates’ work readiness to work competently [5].

Developing competence is an essential component of nursing education to ensure nursing graduates are prepared with the essential knowledge, skills, and attitudes to enter the workforce and functioning safely in entry-level positions [2]. Evaluating competency can help not only educational providers ensure that anyone who completes a degree or course has achieved a required level of competency, it also provides assurance to accrediting and regulatory bodies that completion of an educational endeavour indicates that graduates are competent to practice safe and quality care [2]. However, there is still some discrepancy. A systematic review of nursing graduate competencies reported that only a small proportion of experienced nurses and nurse managers believed that nursing graduates were competent on completion of their undergraduate education [6,7]. This review also discovered tensions in expectations in that the expectation among experienced nurses was that new graduates should be able to provide unsupervised nursing practice whereas others considered it was impossible for nursing graduates to attain complete practice readiness before entering the workforce as the generalist educational preparation [6]. Another study on nurse managers’ perceptions of newly graduated nurses’ performance found that numerous competencies were described as important. However, when it came to the importance of particular competencies, there were differences between work settings, such as hospitals, home health agencies, and nursing homes [7]. Therefore, identifying where educators and employers concur or diverge in their understanding of competency is critical to inform how nurses are prepared for and supported into the workplace and determine what is appropriate with regard to expectations about their practice competency.

Additionally, education providers still need to investigate emerging competencies required for new graduate nurses in order to meet the demand of the contemporary healthcare system. For example, nursing professionals are now expected to play a central and extended role in supporting ageing populations with changing illness patterns in increasingly diverse care delivery environments [8,9]. Furthermore, nursing competencies should be also considered in the global context. As global healthcare issues change competency expectations must change to meet these expectations. For example, competently managing emerging infectious diseases [8]. Notably, although the quantity of publications related to competence has increased few studies have explored competency from a global context. Therefore, regularly reviewing and reflecting on current contexts of competence is essential to inform and drive preparation of nurses whose role will be to ultimately protect the public in a global healthcare environment.

Of note, while competence has been a broadly used concept in nursing, criteria to ex-plain competency have changed over time, reflecting how nurses’ roles are constantly evolving in response to healthcare needs. While discussion about nursing competence has increased exponentially in recent years, debate remains about the definitions of competence and competency, particularly related to new graduate nurses. Both are used inter-changeably and inconsistently in the literature [10]. However, competence is generally de-scribed as dynamic, has a holistic meaning, and refers to capability across multiple dimensions, whereas a competency is a specific capability or an observable ability integrating several components, such as knowledge, skills, attitudes, and values [10,11]. Therefore, in this paper, competence is used to refer to an overall capacity in attaining multiple competencies that integrate certain attributes, such as knowledge, skills, and attitudes as a professional registered nurse.

## 2. Aim

This paper aims to explore how nursing competence and competency in relation to newly graduated nurses have been used in the current global literature; we focused our attention on the current issues relevant to newly graduated nurses’ competence in the published articles. It is hoped that findings will provide important insight into current global status of new graduates’ competence, which can be used by nursing educators and employers to prepare and support new graduate nurses for their practice.

## 3. Methods

### 3.1. Design

Summative content analysis was used to identify trends and relationships in the contextual use of nursing competence within the literature [12]. In a summative approach to content analysis, data analysis begins with searches for occurrences of the identified words by hand or computer. Word frequency counts for each identified term are calculate-ed, which is used to identify patterns in the text and allows for interpretation of the context associated with the use of the words. This approach has certain advantages: it is an unobtrusive and nonreactive way to study the phenomenon of interest. It can also provide basic insights into how words are used [12].

### 3.2. Method

Semantic analysis (whereby meaning is drawn from text) software Leximancer was used for the content analysis. This software efficiently analyses large volumes of textual data and examines it without pre-existing assumptions about the meanings of words, identifying key concepts and how they relate to each other [13]. Competence has been discussed in depth over several decades, therefore, manually reviewing large numbers of documents regarding new graduate nurses’ competence would be difficult, whereas a computer-assisted content analysis can effectively characterise the global features of a large corpus of work and identify relationships between particular concepts and themes [14]. A strength of the Leximancer system as a computer-generated analytic tool is that it provides rigour, stability, and reproducibility [15]; stability is equivalent to inter-coder reliability [16], and reproducibility means that the same result is produced no matter how many times a dataset is coded and recoded [16,17]. However, researchers still need analytical sensitivity and judgement in interpretation because the findings may be limited to the broader meanings in the data [12,15].

### 3.3. Search Process

A search in the computerised databases produced 5103 international studies related to the nursing competence of new graduates. The search terms were ‘graduates’, ‘compe-tence’, and ‘competency’, as well as related terms such as ‘preparedness’, ‘readiness’, and ‘efficacy’. To select probable samples, duplicated articles and references unrelated to nursing competence were manually deleted. A total of 3225 international studies were selected as the probable sample (see Figure 1). The titles and abstracts of the probable samples were collected as indicative evidence of the content of new graduates’ nursing competence because titles and abstracts are lexically dense and focused on the core issues presented in articles [18,19].

### 3.4. Leximancer Program Setting for Data Analysis

After importing the data into Leximancer for analysis, there is some consideration of the program setting. Firstly, the stop-word list was set. Frequently occurring words that hold weak semantic information are called stopwords. These words do not contribute towards the context or information of the documents and should be removed during indexing and before querying by an information retrieval system [20]. Thus, to remove non-lexical or meaningless words and weak semantic information in the analysis, the program has a default setting for the stop-list. Additionally, some general terms used commonly in abstracts, such as ‘study’, ‘article’, ‘purpose’, ‘aim’, ‘background’, ‘method’, ‘conclusion’, ‘data’, and ‘results’, that did not add meaning to the concept maps, were also removed as per Cretchley and colleagues’ suggestion [19].

The data was analysed in two steps. The first step of Lexiamncer was an initial exploratory analysis which provided an overall view of the imported data. This initial analysis showed ‘Nursing’, ‘Health’, and ‘Patients’ as the most frequently mentioned words, but these were deemed too broad to find the important meanings related to nursing competence. Considering the purpose of this analysis, the review used a profiling analysis as a second step to focus on the concept ‘competence/competency’. This method allows the discovery of new concepts that are relevant to it. This was an effective way to zoom in on topics in the data and could be useful in analysing a specific research question or issue in a large text corpus covering a range of themes [20].

A key feature of Leximancer is that the information is displaying a visual map that provides a bird’s eye view representing the main concepts (frequently occurring words) within the text as well as the information about how they are related [20]. As see Figure 2, the themes that contribute to nursing competence as per the heated map relevancies and connectivity order. This means that the hot colours (red and orange) denote the most important themes, and cool colours (green and blue) denote less important themes. Moreover, the connection and the location of each word shows there is possible assiciation between two words [20].

## 4. Findings

Five themes emerged concerning nursing competence and competency related to newly graduated nurses from current global references: (1) standardisation of nursing competencies with emerging competencies areas, (2) assessment to measure competency levels, (3) nursing graduates’ expectations and achievement, (4) safe and quality practice with teamwork, and (5) competence curriculum development (see Figure 2).

### 4.1. Standardisation of Nursing Competencies

Standardisation of nursing competencies was discovered as the most significant theme in the selected literature. Unsurprisingly, the dominant concept in this theme and most frequently mentioned was ’competency’ with 5135 counts. This is because this concept comprised the main topic and the key search terms in the selected references. As seen in the concept map in Figure 2, the concepts’ essential/core’, ‘standards’, and ‘accreditation’ were closely situated to the central concept ‘competency’, which means these concepts are highly associated each other. For instance, ‘competency’ and ‘essential/core’ were mentioned together 720 times and these two words were mentioned together with 54% likelihood. The following examples show the importance of standardised nursing competencies for the preparation of new graduates with essential/core competencies have been often discussed in the current literature.


*Students are expected to graduate with competencies in accordance with professional standards that promote their safe and comprehensive nursing care provision. (#2001)*



*One approach for nursing success is standardizing the entry-level education for nurses and developing a uniform professional development and career advancement trajectory with appropriate incentives to encourage participation. (#2244)*


This analysis also discovered two emerging new competency areas: informatics and cultural competency. Firstly, ‘competency’ was often mentioned with ‘informatics’, at 131 counts with 40% likelihood, and the concepts were closely situated. This can be explained by information technology having become more utilised in the current healthcare system, so informatics skills and abilities were frequently discussed in this context. Additionally, ‘competency’ and ‘cultural’ were mentioned together 214 times with 38% likelihood in the text. This indicates that culturally safe care was also a prominent concept in the data, suggesting that cultural competency had been often considered an essential competency issue in the global context.


*Many health-care organizations and associations recommend that registered nurses be culturally competent and technologically savvy to compete in today’s global society. (#715)*



*The expectation that nurses provide effective care across varied population groups highlights the need for attainment of cultural competency by baccalaureate nursing graduates. Nursing programs must develop strategies to address this educational need. (#855)*


### 4.2. Assessment to Measure Nursing Competencies

‘Assessment’ was one of the predominant concepts discovered in this review. This word was mentioned 2253 times in the text. This concept was highly associated with ‘measurement’ at 22% likelihood. These concepts reflected the importance of assessment to measure nursing competencies in the current context of nursing competence. In addition, ‘self-assessment/self-perceived’ was closely situated and occurred in the international references 476 times. Self-assessment methods could have been widely used to measure nursing competencies with traditional assessment methods. The examples of reviews mentioning assessment methods is as follows.


*The three most common methods used to evaluate competence (direct observation, self-assessment and practice portfolios) lack reliability and validity; the processes are subjective, and assessors may be making judgements on imperfect evidence. (#515)*



*Forty seven nurses commencing a 12-month graduate nurse programme were invited to undertake a self-assessment of their level of competence at four-time points; commencement, 3 months, 6 months and 12 months (#859)*


### 4.3. Nursing Graduates’ Expectations and Achievement

‘Graduates’ was one of the frequently occurring words in the selected international literature (770 counts, 15% relevance). ‘Competency’ was mentioned with ‘graduates’ with 28% likelihood (215 counts), and ‘graduates’ was also often linked with ‘baccalaureate’ (bachelor nursing degree). This can be explained that Nursing Baccalaureate degree graduates may be internationally considered one of the main target groups in nursing education and practice. As seen in Figure 2, the concepts ‘expected’ and ‘achieve’ were closely situated with ‘graduates’ and ‘entry-level’. This could describe many studies about which particular nursing competencies were expected of newly graduated nurses and measurement of their achievement. The most common findings for expected competencies or abilities of graduates were appropriately caring for a diverse population, high-quality ethical and clinical care, and caring for older persons.


*Nursing education should emphasize to a greater extent ethical competency and training for the challenging situations students will encounter in clinical practice. (#1971)*



*The proportion of older adults in the population is rapidly increasing, and this trend is expected to continue. Because more than half of all new graduates eligible to enter the nursing workforce are prepared, it is critical these new nurses are well prepared to care for older adults. (#1319)*


### 4.4. Safe and Quality Professional Nursing Practice with Teamwork

Safe and quality nursing practice was a significant issue related to nursing competency. This theme consisted of ‘safe/safety’ and ‘quality’. The word ‘safe/safety’ was mentioned 854 times in the selected global titles and abstracts. These terms were mainly used for describing patient safety as an outcome of the competent practice of newly graduated nurses. ‘Competency’ was mentioned with ‘teamwork’ with 26% likelihood (59 counts). The concept ‘teamwork’ mainly described the importance of interprofessional collaboration for effective practice.


*Although teamwork and interprofessional collaboration are critical to patient safety, nursing, medical, and allied health graduates often feel ill-prepared to confidently communicate and collaborate with other team members. (#490)*


### 4.5. Competence Curriculum Development

This theme contained ‘curricular’ and ‘framework’. ‘Curricular’ occurred in the selected references 559 times and was mentioned together with ‘competency’ with 23% likelihood. This analysis found a continuous effort to design and redesign nursing curricula with required competencies. Examples mentioning curriculum development were as follows.


*This study has identified the need to develop a standardised competency-based educational and training program for all European countries that will ensure the practice and policies that meet both the standards of care and the broader expectations for professionalisation of the disaster and crisis workforce. (#967)*



*Embedding pain management core competencies into prelicensure nursing education is crucial to ensure that nurses have the essential knowledge and skills to effectively manage pain and to serve as a foundation on which clinical practice skills can be later honed. (#308)*


## 5. Discussion

This paper explored how nursing competence and competency related to new graduates have been used in the current literature. Three main usages were identified. The first was standardisation of nursing competencies for new graduates; education providers preparing nursing students and graduates achieving core nursing competencies as professional registered nurses. This is in line with the finding of Smith and McCarthy [21] that nursing graduates from bachelor’s degree programs should be educated as the largest number of nurses eligible for registered nurse licensure in most employment settings. However, this contrasts with other published studies which indicate that employers might seek a more competent workforce, including new graduate workforce in specialised practice areas, to protect the public and achieve quality practice outcomes [22]. Transitioning to specialty areas could also explain why new graduates frequently reported that they felt unprepared in nursing knowledge and skills for practice after graduation; they were prepared for general nursing rather than specific situations where they would work as registered nurses [23]. Therefore, these findings suggest educators prioritise which core competencies should be focused on during educational preparation that will help develop competent generalist professional nurses. Future employers could then support nursing graduates to improve their competence in specialised areas.

The second finding was that emerging competencies are required in the current healthcare environment. This review discovered culturally safe practice and healthcare informatics as predominant concepts related to nursing competencies in the global context. The importance of the trend in cultural competency is because healthcare professionals are expected to provide quality and safe care for patients from culturally and linguistically diverse backgrounds. Therefore, providing multicultural educational support and training for the nursing workforce to be prepared for culturally competent practice is often cited in the literature [24]. Interestingly, the discussion on cultural competency has expanded to marginalised or smaller groups in the general population, such as LGBTQ+ groups [25,26]. This suggests that further educational strategies should be developed to better prepare nursing graduates for culturally safe practice also to meet the needs of diverse minority populations.

Another growing trend in nursing competency is related to healthcare information technology. This finding may reflect that information technology in a healthcare system has become a key element due to nurses’ roles becoming increasingly reliant on and inter-twined with digital health tools, such as electronic records, mobile computing devices, telehealth, and robotics [27]. A systematic review on informatics competency reported that many informatics competency lists had been developed for entry-level nurses in several countries over the past two decades [27]. Particularly as a result of the COVID-19 pandemic, where there has been an exponential increase in virtual care delivery across the world, further research is needed to evaluate the readiness of new graduates as a healthcare workforce to effectively use digital health tools to support care delivery. Therefore, it is suggested that standardised sets of core competencies related to health and information technology should be included in entry-level nursing education. Additionally, competency standards should be updated to reflect what emerging competencies registered nurses must achieve at the beginning of professional nursing practice to protect the public and ensure quality and safe current practice [28].

Last, assessment to measure levels of nursing competency is another noticeable concept in the global literature. This review found that a common method to evaluate nursing competencies was self-assessment (self-report). This is supported by Cowan’s study [29], who justified a self-assessment method for nursing competencies because it requires critical re-flection of practice, which is positively related to the quality of care, making it a powerful method for assessing competency. However, nursing competence is a complicated concept and needs to be assessed by multiple methods [30]. Furthermore, competencies are acquired and continuously improved throughout education and practice [31]. Therefore, competency assessment methods must be extensively investigated to measure nursing students’ and graduates’ competencies, not only cross-sectionally in comparison to others, but also longitudinally over their program of study and practice. This is critical to ensure graduate nurses achieve at least entry-level competencies in safe and high-quality nursing practice.

This study has a limitation. Although the computational text analysis software allows efficient analysis of large volumes of text data, it still requires analytical sensitivity and judgement in its interpretation. Leximancer provides the benefits of presenting all connections that exist in the data, many of which might not be explored manually, and the risk remains of missing important insights in the findings that may be initially less apparent to the researcher [32]. Therefore, other quantitative and qualitative methods are needed to further explore the topic of this study.

## 6. Conclusions

This content analysis was conducted to explore how nursing competence has been used in the published literature. The relevant titles and abstracts were collected and analysed using Leximancer. The findings were interpreted with relevant concepts. The findings provided important insights into new graduates’ perceived competencies and highlighted that regularly revising standardised nursing education for entry-level practice and nursing graduates is essential. This, in turn, reflects the demand for cultural and healthcare informatics technology competency in the current healthcare environment. This analysis has also raised important questions about competency assessment. Although self-assessment is common for evaluating new graduate nurses’ competency levels, further research should investigate and develop assessment methods to measure multiple aspects of nursing competence throughout education and practice.

## Figures and Tables

**Figure 1 healthcare-10-01071-f001:**
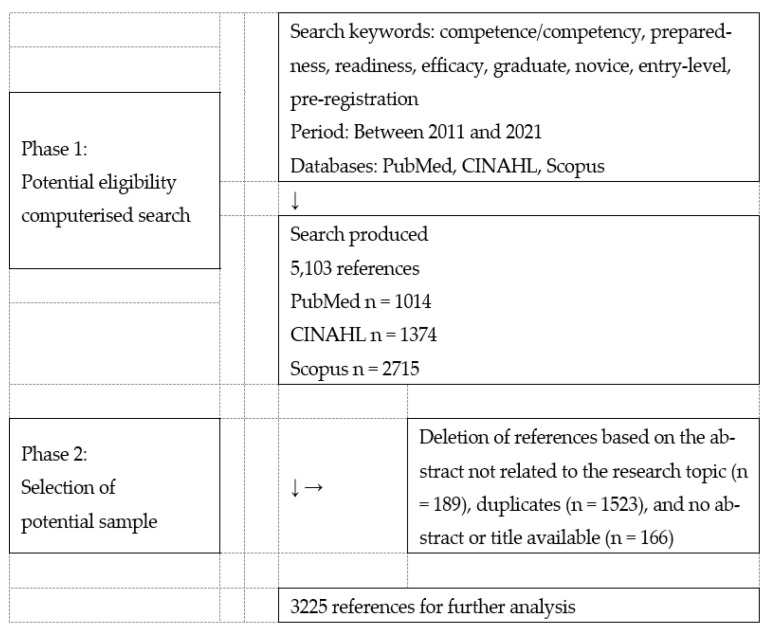
Literature search process.

**Figure 2 healthcare-10-01071-f002:**
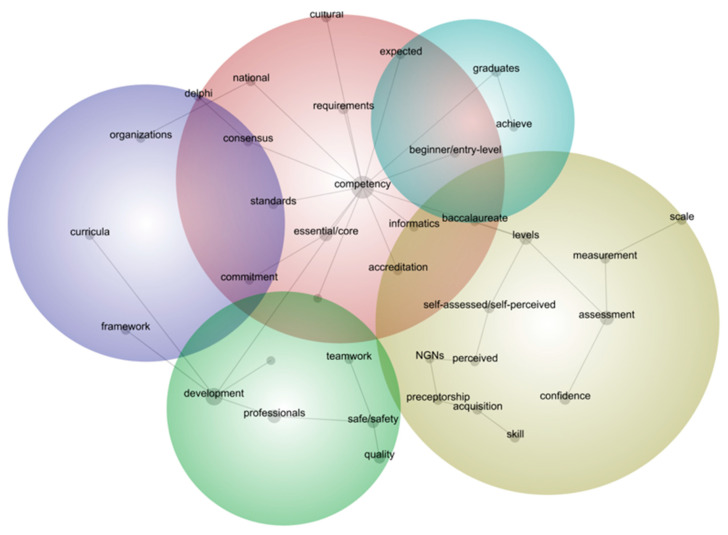
Concept map of ‘competence/competency’ profiling in the global literature.

## Data Availability

Not applicable.

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
