# Peer review of "The Current Contexts of Newly Graduated Nurses’ Competence: A Content Analysis"

_healthcare, 2022, doi:10.3390/healthcare10061071_

Round 1

Reviewer 1 Report

I fully agree with the importance of having a competent nursing workforce over all interesting study.

It would be the subject of an other study over ever it could be interesting to compare the curicula of nurse training in different comparable contexts like Canada, North america , Europe (although in Europe it be be different from one country to an other).

Presentation of results 4.1 would it be possible to make a graph with the results?

In the discussion it could be interesting to dicuss the advance practice nurses as it does exist in Canada

In the discussion, e. health and information technology is not (yet) part of the nurse training . Should it be included in the curicula?

Author Response

Thank you for giving us the opportunity to submit a revised draft of our manuscript titled ‘The current contexts of newly graduated nurses’ competence: a content analysis’ for publication in the Healthcare.

We appreciate the time and effort that the reviewers have provided valuable and insightful comments on this manuscript. We have been able to incorporate changes to reflect on the suggestions provided by the reviewers.

We look forward to hearing from you in due time regarding our submission and to responding to any further comments you may have.

Thank you. 

Reviewer 2 Report

The study used the content analysis to analysis the data from the last ten years of Pub-Med, CINAHL and Scopus database. A search in the databases produced 5,103 international studies related 91 to the nursing competence of new graduates. A total of 3,225 titles and abstracts regarding nursing competence were identified and analyzed using the big-data analysis software Leximancer. The current study has merits and contributes to the existing knowledge in its current format. However, it needs to incorporate some changes. I recommend the author(s) make the suggested changes indicated below:

1.     Although the Introduction section mentioned the purpose of the research, there were no clear research questions. It is suggested that additional explanation should be provided. For example: What are the “practical” issues that need to be resolved? Therefore, the motivation for the paper is unclear. I'm not sure what problem the paper is meant to address. It does outline a situation that the authors would like to explore, but there is no real research gap identified or contributions that might arise from studying this problem.

2.     In order to make a clear contribution to the literature, I urge the authors to make a greater contribution, that is grounded in a well-though “theoretical background”: The current description of the research background is still too simple, and the literature should be updated to 2021 or 2022.

3.     It is not easy to understand the meaning of Figure 1, the authors may be able to provide more explanation.

4.     Please confirm the reference format according to the requirements of the journal.

5.     Research findings should strengthen explained clearly, especially by cite some examples. It will be easier to understand this part.

Author Response

(The authors gave the same response as above.)

Round 2

Reviewer 2 Report

Thank you for making the appropriate corrections to my comments.

The manuscript has been much improved and is in a nice condition now.

I considered that the modifications made improve the quality of the manuscript.